# *Staphylococcus* spp. Causatives of Infections and Carrier of *blaZ*, *femA*, and *mecA* Genes Associated with Resistance

**DOI:** 10.3390/antibiotics12040671

**Published:** 2023-03-29

**Authors:** Laryssa Ketelyn Lima Pimenta, Carolina Andrade Rodrigues, Arlindo Rodrigues Galvão Filho, Clarimar José Coelho, Viviane Goes, Mariely Estrela, Priscila de Souza, Melissa Ameloti Gomes Avelino, José Daniel Gonçalves Vieira, Lilian Carneiro

**Affiliations:** 1Biotechnology Department, Medicine Tropical and Health Public Institute, Universidade Federal de Goiás, Goiania 74605-020, Brazil; 2Colemar Natal e Silva Camp, Biotechnology Department, Medicine Faculty, Universidade Federal de Goiás, Goiania 74605-020, Brazil; 3Samambaia Camp, Engineering Department, Engineering School, Universidade Federal de Goiás, Goiania 74690-900, Brazil; 4Computer Department, College of Computer Engineering, Pontifíca Universidade Católica de Goiás, Goiania 74605-020, Brazil; 5Inovation Department, Instituto de Biologia Molecular do Paraná, Curitiba 81350-010, Brazil

**Keywords:** hospital infections, qPCR, resistance, *Staphylococcus*, penicillin G, oxacillin

## Abstract

*Staphylococcus* spp. have been associated with cases of healthcare associated infections due to their high incidence in isolates from the hospital environment and their ability to cause infections in immunocompromised patients; synthesize biofilms on medical instruments, in the case of negative coagulase species; and change in genetic material, thus making it possible to disseminate genes that code for the acquisition of resistance mechanisms against the action of antibiotics. This study evaluated the presence of *blaZ*, *femA*, and *mecA* chromosomal and plasmid genes of *Staphylococcus* spp. using the qPCR technique. The results were associated with the phenotypic expression of resistance to oxacillin and penicillin G. We found that the chromosomal *femA* gene was present in a greater proportion in *S. intermedius* when compared with the other species analyzed, while the plasmid-borne *mecA* gene was prevalent in the *S. aureus* samples. The binary logistic regression performed to verify the association among the expression of the genes analyzed and the acquisition of resistance to oxacillin and penicillin G were not significant in any of the analyses, *p* > 0.05.

## 1. Introduction

Healthcare-associated infections (HAIs) have been one of the main causes of morbidity and mortality not only in Brazil but worldwide, with a global estimate of 51.4% of morbidity and mortality in intensive care units (ICUs) [1]; about 3% to 15% of hospitalized patients in Brazil acquire nosocomial infections [2]. Such infections are transmitted during the provision of health care, and may occur during hospitalization or after discharge, when related to the procedures performed [3]. HAIs are related to biosafety failures such as the incorrect use of PPE (personal protection equipment) and EPCs (collective protection equipment) [4]. HAI cases are also associated with the patient’s long hospital stay, immunosuppression status, severity of the initial disease, inappropriate use of antibiotics [5] and surgical procedures [6].

For HAI control, much has been said about the importance of proper hand hygiene by health professionals—especially within ICUs, the most conducive place for the spread of pathogens. Correct hand hygiene by health professionals is essential to minimize cases of cross-transmission between patients. This transmission occurs when there are pathogens colonizing the skin of patients or hospital objects and the health professional is contaminated through direct contact with these surfaces. Thus, if there is no hand hygiene at the appropriate times and in the advisable way, this professional will disseminate the pathogens carried on their person when offering assistance to other patients [7].

The main etiological agents of HAIs comprise certain bacterial species of the genus *Staphylococcus* [8]. *Staphylococcus* species are a genus of Gram-positive bacteria characterized by the presence of the enzyme catalase, considered a virulence factor, since its function consists of the degradation of H_2_O_2_ into O_2_ + H_2_O and prevention of neutrophils from causing bacterial cell death [9].

Staphylococci are divided into two groups: producers and non-producers of the enzyme coagulase. In the coagulase-positive group, the species *S. aureus* stands out, which makes up the residual microbiota of the nasal passages of about 20% of the population, in addition to the individuals who occasionally carry it. This species is of great importance due to its high capacity to cause infections and adapt to different environmental conditions [10].

*S. aureus* is one of the main causes of healthcare-associated infections (HAIs) and, currently, of infections acquired in the community, resulting in serious consequences. This pathogen is involved in infections of the bloodstream, skin, soft tissues, and respiratory tract and can trigger diseases from simple to serious in nature such as pimples, boils, cellulitis, pneumonia, meningitis, endocarditis, and sepsis, among others [8]. Furthermore, strains of *S. aureus* are also related to methicillin resistance [11].

Coagulase-negative staphylococci are present on the surface of the skin and constitute about 90% of the body’s normal microbiota. They become pathogenic when the skin is broken or with the use of medical devices such as catheters. Infection through medical devices occurs due to the bacterial capacity to create biofilms on these surfaces, these biofilms serve as protection, since they are not susceptible to the action of disinfectants [12].

The species *S. epidermidis* and *S. haemolyticus* are related to nosocomial infections and are the most prevalent coagulase-negative species in individuals [13,14], being responsible for most nosocomial infections such as bacteremia and infective endocarditis [15], with the ability to synthesize biofilms on medical devices such as prosthetic devices and intravenous catheters [16]. Another important factor is that staphylococci are considered a reservoir of antimicrobial resistance genes due to their ability to exchange genetic material with other species [17].

Staphylococci have rapidly evolved to show resistance to beta-lactams: penicillins, cephalosporins, carbapenems, and monobactams [13]. The genes that code for resistance mechanisms can be located on plasmids and/or on the chromosome depending on the way that such a gene was obtained. Staphylococci have two resistance mechanisms that stand out: the production of beta-lactamases and the production of PBP2a [18].

The production of beta-lactamases, extracellular enzymes, occurs through the expression of the *blaZ* gene, usually located in the genomic material; however, it can also be present on plasmids. Beta-lactamase inhibits the action of beta-lactams by cleaving the beta-lactam ring, which provides the mechanism of action for this class of antimicrobials. The *blaZ* gene is regulated by two other genes, the *bla*R1 antirepressor and the *bla*I repressor. After exposure to beta-lactams, *bla*R1, a transmembrane sensor-transducer, undergoes autocatalytic cleavage, promoting cleavage of the repressor gene, *bla*I, thus allowing transcription of *blaZ* [18,19].

The PBP2a or PBP2’ proteins produced from the expression of the *mecA* gene are fully functional for the bacterial cell but have a low affinity for beta-lactams [20]. Expression of the *mecA* gene is constitutive or induced by beta-lactam antibiotics. The *mecA* gene is inserted into the staphylococcal chromosome, through a mobile genetic element called staphylococcal chromosomal cassette mec (SCCmec). The SCCmec is composed of several essential genetic elements: the mec complex, composed of the IS431 pathogenicity island, the *mecA* genes and their regulators *mecI* and *mecR1*, and the ccr complex (chromosome recombinase cassette), characterized by the presence of genes that encode recombinases. In all types of SCCmec, the *mecA* gene sequence is highly conserved in strains of S*. aureus* and *Staphylococcus* spp. negative coagulase [18].

In addition to the product encoded by the expression of the *mecA* gene that can induce methicillin resistance in *S. aureus*, another gene may also be associated with the resistance level of these bacteria, the *femA* gene (essential factor for the expression of methicillin resistance). The *femA* gene produces a protein of a size of 48 kDa that can restore methicillin resistance in *S. aureus* [21].

The acquisition of resistance mechanisms by bacteria in the hospital environment has made the treatment of HAIs difficult, and has increased the morbidity and mortality rates of patients in a state of immunosuppression. Therefore, the objective of the present work is to investigate the *blaZ*, *femA*, and *mecA* resistance genes in bacterial samples isolated through material collected from the hands of ICU health professionals, since the hands are the main vector of cross-transmission of pathogens.

## 2. Results

In this study, 48 isolates of *Staphylococcus* spp. were obtained, and approximately 90% of them were classified as coagulase negative. The predominant species isolated from the material coming from the hands of health professionals in an intensive care unit (ICU) was *S. haemolyticus* (20 isolated), followed by *S. intermedius* (7 isolated), and the least common species were *S. pasteuri* and *S. epidermidis* (1 each isolated) (Figure 1).

Among the isolates tested for the *blaZ* gene of the chromosomal material, none of the samples, regardless of species, was positive. On the other hand, the *femA* gene was widespread among the isolates, being present in 100% of *S. intermedius*, *S. epidermidis*, *S. carnosus* subsp. *utilis*, and *S. pasteuri* tested; and positive in 50% of *S. carnosus* subsp. *carnosus* and *S. auricularis* and 30% of *S. haemolyticus* samples.

However, none of the *S. agnetis* and *S. aureus* subsp. *aureus* were positive for chromosomal *femA*. As for *mecA*, a single isolate of *S. haemolyticus* tested positive for the chromosomal gene (Table 1). The *femA* gene showed a higher proportion in *S. intermedius* when compared with *S. haemolyticus*, *p =* 0.003. When compared with all *Staphylococcus* species, the result showed a higher proportion in *S. intermedius*, *p =* 0.004; thus, the *S. intermedius* species is associated with the *femA* chromosomal gene.

Regarding the analyzed plasmid DNA, the *blaZ* gene was present in 100% of *S. agnetis* and absent in *S. aureus* subsp. *aureus*, *S. epidermidis*, and *S. pasteuri*. The *femA* was present in *S. auricularis* (100%) and absent in *S. agnetis*, *S. epidermidis*, and *S. pasteuri*. The *mecA* gene was present in 100% of the *S. agnetis* tests and absent in *S. haemolyticus*, *S. auricularis*, *S. epidermidis*, and *S. pasteuri* (Table 1).

When considering the plasmid, the *femA* gene showed no difference in proportion of presence. When considering the comparison among *S. haemolyticus* and *S. intermedius*, *p =* 0.47; likewise, when comparing *S. intermedius* with all species, *p =* 0.41. However, the *mecA* gene showed a proportionally higher presence in *S. aureus* subsp. *aureus* when compared with other species, *p =* 0.04.

Results previously published by our group [22] showed that 39.4% and 42.4% of *S. haemolyticus* species were resistant to penicillin G and oxacillin, respectively. *S. auricularis* exhibited values of 15.2% for both tested antibiotics and *S. intermedius* of 12.1% for both antibiotics. Therefore, in this study, we associated species that were positive for the tested genes and that showed resistance to these antibiotics in the susceptibility test in the previous study in order to relate resistance to the acquisition of resistance mechanisms expressed by such genes, which act mainly in the synthesis of the bacterial cell wall.

Table 2 presents the data obtained. Of the *S. haemolyticus* isolates, two strains expressed from chromosomal *femA*, from chromosomal *mecA* (1), and from plasmid *femA* (2) and together with the expression of these genes also showed resistance to oxacillin (Table 2).

From the following isolates, only one chromosomal *femA* resistant gene for penicillin was found: S*. intermedius* (3) and *S. pasteuri* (1) (Table 3). Despite considering each sample and gene separately, some isolates were positive for more than one gene. In the Appendix A, it is possible to observe the identification of each sample and its results for the presence of genes and positivity for isolated antibiotics—in addition to being able to visualize the samples that were not tested for some of the genes, due to the lack of growth at the time of their reactivation.

The binary logistic regression performed to verify the association between the presence of the investigated genes and the expression of resistance to the antibiotics penicillin G and oxacillin was not significant in any of the analyses *p* > 0.05 (Appendix A).

Therefore, X^2^ statistics were used for the association between two variables, verifying whether resistance to penicillin G or oxacillin in the genus *Staphylococcus* could be associated with the presence of one of the researched genes. There was no association between the presence of *blaZ femA*, or *mecA* genes (plasmid or chromosomal) and resistance to the antimicrobials penicillin G or oxacillin, *p* > 0.05 for all analyses. For the *blaZ* and *mecA* (chromosomal) genes, it was not possible to perform the association test due to low or no variables in the model. All results of the X^2^ association tests can be seen in Appendix A.

## 3. Discussion

Among the staphylococci, the species *S. aureus* stands out in the genus due to its strong relationship both with nosocomial infections and antimicrobial resistance. Even at the beginning of the era of antibiotics, it was possible to identify the first strain resistant to penicillin, with the action of the *blaZ* gene that codes for β-lactamases; these enzymes cleave the β-lactam ring, preventing the action of the antibiotic. Initially, this was a problem found only in the hospital environment, but it is already possible to find strains of multidrug-resistant *S. aureus* in infections in the community.

Currently, both coagulase-positive and negative community staphylococci have a high rate of resistance to penicillin G [23]. In the present study, we obtained four *S. aureus* isolates. Of these samples, two were positive for the *femA* plasmid and showed resistance to penicillin G in the susceptibility test. One of the samples was positive for both the *femA* and *mecA* plasmid genes but remained sensitive to oxacillin. This finding may be related to the presence of silent genes that are carried in the bacterial genetic material but are not expressed and, therefore, there is no phenotypic resistance. This inactivation and activation of the gene can occur due to different conditions such as the culture medium [24]. Another factor that can interfere with phenotypic resistance is polygenic resistance [25].

*S. haemolyticus* and *S. epidermidis* have been the most frequently isolated staphylococci in hospital environments and are mainly associated with medical devices [26]. In one study [27], there was the identification of 11 isolates of coagulase-negative staphylococci, being 45% *S. epidermidis* and 27% *S. haemolyticus*. Of 200 strains of coagulase-negative staphylococci obtained from patients with nosocomial bacteremia in Turkey, 87 corresponded to *S. epidermidis* and 23 to *S. haemolyticus*, both of which are more prevalent among 12 other species also isolated [28].

Other studies also reported similar data [29], where 41% of isolates from ICU patients in Sri Lanka were identified as *S. haemolyticus*, with the total number of coagulase-negative staphylococcus (CNS) equal to 82 samples. Researchers [30] in Nepal obtained 123 CNS isolates from hospital materials in the ICU and wards; in this case, the most common species was *S. epidermidis* (42%), and *S. haemolyticus* represented 9% of isolates. Another study [31] collected nasal and hand samples from 125 health professionals from a hospital in Nepal. Of the 250 samples, 203 were identified as SCN and 38 as *S. aureus*.

According to the data, in our study, 41.6% of the isolates were *S. haemolyticus*, and in none of these was the *blaZ* chromosomal gene found. However, one sample was positive for chromosomal *mecA* and the *femA* plasmid and showed resistance to oxacillin, while samples positive for *femA* (plasmid or chromosomal) were resistant to penicillin G. In further consideration of the *S. haemolyticus* species, one of the isolates was positive for *blaZ* and plasmid *femA* genes; however, it did not show positive test for resistance to the two analyzed antibiotics.

*S. intermedius* took second place with approximately 14% of the samples. In one of the isolates, *femA* was found in chromosomal and plasmidial DNA, but the isolate did not show resistance to oxacillin or penicillin G. In contrast, a negative sample for the chromosomal genes *blaZ* and *femA* and for the plasmidial *mecA* was resistant to penicillin G. However, the chromosomal genes *femA* and *mecA* were not investigated and, therefore, the acquisition of resistance may be related to the presence of these genes, since samples that were positive for chromosomal *femA* alone or together with plasmid *femA* showed resistance to penicillin G.

The species *S. agnetis* represented 6% of the total samples isolated in this study. Despite not being significant in the area of human health, these staphylococci have been reported due to their veterinary importance as a potential etiological agent of infections in birds such as broilers and cattle, both of which have great commercial value [32,33]. As shown in the results section, *S. agnetis* tested positive for the plasmidial *blaZ* and *mecA* genes and showed resistance to both oxacillin and penicillin G in the susceptibility test. Suggesting that this species may share resistance genes with other species in the hospital environment, as well as hinder the treatment of infected birds and cattle.

*S. carnosus* was only classified as belonging to the genus of staphylococci in 1982 [34]; however, it has been used since 1950 to ferment sausages and is a non-pathogenic microorganism. It is considered as a contributor for food quality, providing flavor and fermentation control [35]. In the present study, when we consider all samples of *S. carnosus*, we have the presence of all the genes of the plasmidial material studied, while in the chromosomal DNA, only *femA* was present. However, none of the positive samples for such genes showed resistance to the tested antibiotics. This condition suggests the possibility that these strains, even without the potential to cause human infections, transfer such resistance genes to other species of greater clinical importance, increasing the difficulty in treating patients.

Even today, there are few reports on infections caused by *S. pasteuri*, but it is already known that it infects the gastrointestinal microbiota of children with active celiac disease and that it is a contaminant of platelet transfusions, with infective endocarditis [36]. The isolate from this study showed expression of the chromosomal *femA* gene and resistance to penicillin G but remained sensitive to oxacillin.

*S. auricularis* was discovered in 1983. It colonizes the external auditory canal, it has been the cause of rare community and nosocomial infections, and its relationship with prosthetic valve endocarditis was recently described [37]. Furthermore, this species was isolated from 10.7% of a total of 5447 low-birth-weight infants who participated in a survey that associated *S. auricularis* infection as a cause of early onset sepsis. According to an antibiogram test, only 17% of the isolated strains were susceptible to penicillin [38]. Here, we analyzed 6 isolates of *S. auricularis*, one sample was positive for *blaZ* and *femA* (plasmidial) and showed resistance to the two antimicrobials analyzed. Other samples were also resistant to oxacillin and penicillin G but it was not possible to investigate the presence of resistance genes.

In turn, *S. epidermidis* has been considered an opportunistic pathogen of medical relevance due to its ability to produce biofilms and act as a reservoir of resistance genes, rendering it the main cause of nosocomial infections [39], along with *S. aureus* causing infections in both orthopedic and breast catheters and implants [40]. As a result of its high rate of antimicrobial resistance, infections are difficult to treat. In this study, *S. epidermidis* corresponded to 2% of the isolates from the hands of ICU professionals, and this sample was positive for chromosomal *femA* and showed resistance to penicillin G and oxacillin. However, for this sample, only the chromosomal and plasmid *femA* gene was investigated, the latter being negative in the analyses.

One study [41] isolated and identified 12 strains of *S. epidermidis* from 150 smear samples and tracked the *blaZ* and *mecA* genes in 91.7% of the studied samples; in addition to these genes, this same author investigated the presence of other genes that appeared with lesser incidence and 75% of the samples were resistant to penicillin and 66.7% to oxacillin.

Widely described in the literature, the *mecA* gene was present in only one chromosomal DNA sample from *S. haemolyticus* in the present study. Researchers [42] have described that of 129 isolates from the ICU and from health professionals, 86% were identified as *S. aureus* and 20% were positive for MRSA, highlighting the influence of health professionals in the cross-transmission of infections. In another study, detection of *mecA* was reported in 70% (n = 87) of blood, endotracheal tube, and central venous catheter isolates [30], reaching a total of 41% (n = 34) in *S. haemolyticus* isolated from patients with catheter-related bloodstream infection and colonized central venous catheter [29].

In another study [43], the *mecA* gene was isolated in 95% of samples from different surfaces of an ICU bed, and all isolates grew in MRSA medium, even though none of them were of the species *S. aureus*; based on sensitivity tests, it was found that 100% of the samples were resistant to oxacillin.

In one work [44], 15 CNS isolates were identified, belonging to *S. epidermidis*, *S. haemolyticus*, *S. warningeri*, *S. hominis*, and *S. capitis*. The researchers found that 13 strains were resistant to oxacillin and β-lactams. Molecular tests of all samples were positive for the *mecA* gene. Another study [45] reported that of 89 SCN isolates, 63 had the *mecA* gene and all isolates were resistant to penicillin and other antibiotics in different proportions.

The association among the isolation of *blaZ*, *femA*, and *mecA* genes with the acquisition of mechanisms of resistance, mainly against β-lactams, is already known and widely reported. In the present study, statistical evaluation was carried out to relate these variables, the results obtained did not attest to such associations in any of the analyses performed (*p* > 0.05). It is suggested that the small number of samples and events in each of the analyzed variables has negatively interfered in the statistical model.

In our study, some events were null and most of them were less than ten (Appendix A). Furthermore, when performing the X^2^ statistic for association among these variables, verifying whether resistance to penicillin G or oxacillin of the *Staphylococcus* spp. may be associated with one of the genes investigated, no association was found among the presence of the genes *blaZ*, *femA*, *mecA*, and *blaZ* (chromosomal and plasmid) and resistance to the antimicrobials penicillin G and oxacillin, *p* > 0.05 for all analyses.

The CDC (Centers for Disease Control) reports that with the use of preventive measures such as the correct hygiene of hands and the hospital environment, it would be possible to avoid 20 to 30% of HAIs. However, adherence to HAI prevention measures by health professionals is still very low. The incidence of HAI cases associated with increased resistance to antimicrobials by infecting pathogens has further aggravated the situation, thus becoming a global public health problem [46].

The acquisition of resistance mechanisms by bacteria has been aggravated, in part, by the indiscriminate use of antimicrobials; approximately 90% of patients who acquire such drugs use them in a period equal to or less than 3 days. About 50% of antimicrobials are prescribed inappropriately, and in some countries the purchase of these drugs without a prescription reaches around 2/3. In addition to the increase in bacterial resistance achieved by the incorrect use of antimicrobials, this error can lead to other problems such as an increase in more serious diseases, an increased risk of complications, an increase in mortality, and an increase in health costs [47].

Another factor is the difficulty in developing new antibiotics. The production process of a drug is very long, about 10 years. Furthermore, this includes a high financial investment in research and there is no guarantee of developing a molecule that is effective for the treatment of infections and viable for commercialization. Another reason that has discouraged investors is the possibility that a new molecule launched on the market may become infeasible for use in the first few years due to the rapid acquisition of resistance by bacteria. Due to these limitations, the pharmaceutical industry has invested less and less in this field, prioritizing the production of drugs for the treatment of chronic diseases. Therefore, it is necessary to maintain supervision, sales control, prescription, and correct use; otherwise, cases of HAI associated with resistant and multidrug-resistant species will become increasingly recurrent and severe [48].

Based on published and obtained data, the hands of health professionals can be vehicles for cross-transmission of pathogens between immunosuppressed patients, increasing cases of HAI. These pathogens, in turn, can carry genotypic resistance, resulting or not resulting in phenotypic resistance to antibiotics commonly used to treat infections, thus bringing more complications to public health.

## 4. Materials and Methods

The isolates used in this study were collected, identified, and tested for antibiotic susceptibility in the study conducted by Rodrigues and collaborators [22].

The sample collection took place in a public hospital of high complexity specialized in traumatology, in urgent and emergency care in the city of Goiânia, capital of the State of Goiás. This hospital has multi and interdisciplinary teams, with 470 beds for the Unified Health System (SUS), among which, 57 beds are in the ICU, where the samples were collected (05/19 to 09/19), for bacteriological identification in the laboratory.

Volunteers for data collection were nursing professionals such as technicians, nurses, and physiotherapists; over 18 years old; and who agreed to participate in the research. Those nursing professionals who did not agree to participate in the research were excluded from the research. Samples were collected from eight health professionals, and from each professional, samples were collected from the left and right hands, isolating 48 bacteria.

After submission to the Teaching and Research Ethics Committee of the Hospital de Urgências de Goiânia and approval of the study, in accordance with current legislation in Brazil, the Free and Informed Consent Form (TCLE) was applied to health professionals who volunteered to participate in this study. The collection was standardized by approaching the employee at random and inviting them to the survey. The research was approved on 10 May 2019 with CAAE: 08689018.6.0000.0033.

### 4.1. Data Collection

The hands of the professionals who participated in the research were placed separately in a sterile plastic bag containing distilled water and the fingertips were rubbed together for five minutes. At the end of the collections, the samples were placed in an isothermal box and transported to the Microorganism Biotechnology Laboratory (LBMIC/IPTSP) for microbiological analysis.

### 4.2. Bacterial Identification

A volume of 1 mL was pipetted for each sample, the material was inoculated in BHI broth (brain heart infusion), and then these media were incubated at 37 °C for 24 h. The samples that showed turbidity were seeded by exhaustion with three different types of culture media in Petri dishes: blood agar (AS) for the identification of *Acinetobacter* spp., MacConkey agar (AMC) for the identification of *Escherichia coli* and other enterobacteria, and agar salted mannitol (AMS) for the identification of *Staphylococcus* spp., before incubation at 37 °C for 24 h. After the growth of the colonies, isolation and observation of the macro and microscopic aspects of the colonies followed. Another step used was the Gram stain.

As an additional exam to the Gram-staining result, the KOH (potassium hydroxide) test was performed. For identification, catalase and oxidase tests were performed. After confirming the growth and morphology by the Gram technique, those microorganisms with positive catalase tests were seeded again and isolated in salty mannitol agar for *S. aureus* research. For the isolates that presented negative catalase tests, we proceeded to seeding in blood agar, with the purpose of verifying *Streptococcus* spp., *Micrococcus* spp., and negative *Staphylococcus* coagulase.

After the analyses, all samples were kept in a freezer at −80 °C in tubes containing casein-soy broth and 10% glycerol until the moment of reactivation for the molecular analysis of the present study. Thus, in order to complement the results previously published by our group [22], we evaluated the molecular expression of the *blaZ*, *femA*, and *mecA* genes that are associated with the acquisition of resistance mechanisms against the action of beta-lactams.

### 4.3. DNA Extraction and Quantification

For molecular evaluation, the samples were seeded in BHI medium and incubated at 37 °C for 24 h. At this stage, some samples did not show growth. For the viable samples, the extraction of plasmid and chromosomal DNA was performed separately, following the instructions for the Pharmacia^®^ FLEXIPREP (Stockholm, Sweden) and Qiagen^®^ DNA minikit extraction kit (Venlo, Netherlands), respectively.

After carrying out the DNA extraction protocol, the genetic material was quantified using the Thermo Scientific NanoDrop^®^ 2000 Spectrophotometer (Waltham, MA, USA). The quantification was performed individually and sterile milliQ water was used as a blank for proper calibration between the quantifications of each sample.

### 4.4. qPCR

The identified samples were subjected to the characterization of resistance genes through real-time PCR assays with the Sybr Green Real Time PCR kit (Sybr Green qPCR master mix LOW ROX—100 reactions × 25 µL), adding DNA and specific primers for amplification of each gene. For the internal control, a C protein reactive specific primer was used; for the positive control of the reaction, primers were used to amplify the 16S RNA; and for the negative control, water was added instead of DNA.

The conditions of the real time PCR technique were standardized following the manufacturer’s instructions. A cycling protocol was used under the following conditions: initial denaturation at 95 °C for 2 min; maintenance of denaturation at 95 °C for 15 s; annealing and extension of the oligonucleotides at 60 °C for 60 s; melting curve at 65 °C for 30 s.

For the amplification of *Staphylococcus* spp., the following primers were used: *mecA* (310 bp), (forward) 5′ GTA GAA ATG ACT GAA CGT CCG ATAA3′ and (reverse) 5′ CCA ATT CCA CAT TGT TTC GGT CTA A 3′ [49]; *blaZ* (228 bp), (forward) 5′ AAG AGA TTT GCC TAT GCT TC 3′ and (reverse) 5′ GCT TGA CCA CTT TTA TCA GC 3′ (Araújo, 2019); *femA* (forward) 5′ AAAAAAGCACATAACAAGCG 3′ and (reverse) 5′ GATAAAGAAGAAACCAGCAG 3′ [50].

### 4.5. Statistical Analysis

For association of *Staphylococcus* spp. with resistance genes, a comparison among species was performed, considering the binomial test for two proportions. Statistical analyses were performed using the BioEstat 5.3 software, considering a significance limit of 5%.

OR was calculated by means of binary logistic regression, in order to estimate the probability of the outcome of resistance to the tested antibiotic or sensitivity of the sample analyzed. For regression, the predictors were *blaZ*, *femA*, and *mecA* (chromosomal and plasmid), and the outcome was resistance to penicillin G or oxacillin. Regression was performed for all grouped and individual species. Pearson’s chi-square (X^2^) test [51] for association was calculated for all species grouped, considering genes and resistance to antibiotics already described All confidence intervals were 95% (95% CI) and *p* < 0.05 was considered statistically significant. Statistical analysis was performed using Minitab^®^ software version 19.

## Figures and Tables

**Figure 1 antibiotics-12-00671-f001:**
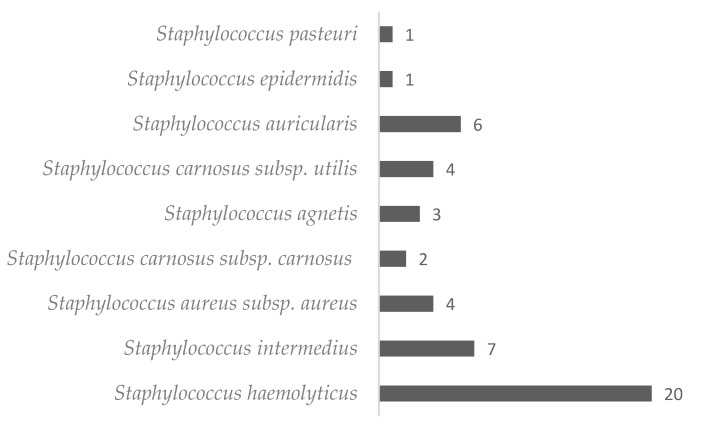
Number and species of *Staphylococci* isolated from the hands of healthcare professionals.

**Table 1 antibiotics-12-00671-t001:** Proportion of the presence of *blaZ*, *femA*, and *mecA* genes in the chromosomal and plasmid DNA of *Staphylococcus* spp. tested.

			*S. haemolyticus*	*S. intermedius*	*S. aureus* subsp. *aureus*	*S. carnosus* subsp. *carnosus*	*S. agnetis*	*S. auricularis*	*S. epidermidis*	*S. carnosus* subsp. *utilis*	*S. pasteuri*
Chromosomal DNA	*blaZ*	Total	12	7	3	2	2	2	0	2	1
		Positive	0	0	0	0	0	0	0	0	0
		*%*	0	0	0	0	0	0	0	0	0
	*femA*	Total	10	6	3	2	1	2	1	1	1
		Positive	3	6	0	1	0	1	1	1	1
		*%*	30	100	0	50	0	50	100	100	100
	*mecA*	Total	13	5	3	1	2	4	0	3	1
		Positive	1	0	0	0	0	0	0	0	0
		*%*	7.7	0	0	0	0	0	0	0	0
Plasmid DNA	*blaZ*	Total	10	7	2	2	1	2	1	3	1
		Positive	1	1	0	1	1	1	0	1	0
		*%*	10	14.3	0	50	100	50	0	33.3	0
	*femA*	Total	12	7	3	2	0	1	0	3	1
		Positive	5	3	1	1	0	1	0	2	0
		*%*	41.6	42.5	33.3	50	0	100	0	66.6	0
	*mecA*	Total	12	7	3	2	1	2	0	3	1
		Positive	0	1	2	1	1	0	0	1	0
		*%*	0	14.3	66.6	50	100	0	0	33.3	0

**Table 2 antibiotics-12-00671-t002:** Isolates resistant to oxacillin and positive for the genes *blaZ*, *femA*, and *mecA*.

	Chromosomal						Plasmid					
Species	*blaZ*	*%*	*femA*	*%*	*mecA*	*%*	*blaZ*	*%*	*femA*	*%*	*mecA*	*%*
*S. haemolyticus*	0	0	2	66.6	1	100	0	0	2	40	0	0
*S. intermedius*	0	0	4	66.6	0	0	1	100	2	66.6	1	100
*S. aureus* subsp. *aureus*	0	0	0	0	0	0	0	0	0	0	0	0
*S. carnosus* subsp. *carnosus*	0	0	0	0	0	0	0	0	0	0	0	0
*S. agnetis*	0	0	0	0	0	0	1	100	0	0	1	100
*S. auricularis*	0	0	1	100	0	0	1	100	1	100	0	0
*S. epidermidis*	0	0	1	100	0	0	0	0	0	0	0	0
*S. carnosus* subsp. *utilis*	0	0	1	100	0	0	1	100	2	100	1	100
*S. pasteuri*	0	0	0	0	0	0	0	0	0	0	0	0

**Table 3 antibiotics-12-00671-t003:** Isolates resistant to penicillin G and positive for the genes *blaZ*, *femA*, and *mecA*.

	Chromosomal						Plasmid					
Species	*blaZ*	*%*	*femA*	*%*	*mecA*	*%*	*blaZ*	*%*	*femA*	*%*	*mecA*	*%*
*S. haemolyticus*	0	0	2	66.6	0	0	0	0	2	40	0	0
*S. intermedius*	0	0	3	50	0	0	1	100	2	66.6	0	0
*S. aureus* subsp. *aureus*	0	0	0	0	0	0	0	0	1	100	2	100
*S. carnosus* subsp. *carnosus*	0	0	0	0	0	0	0	0	0	0	0	0
*S. agnetis*	0	0	0	0	0	0	1	100	0	0	1	100
*S. auricularis*	0	0	1	100	0	0	1	100	1	100	0	0
*S. epidermidis*	0	0	1	100	0	0	0	0	0	0	0	0
*S. carnosus* subsp. *utilis*	0	0	1	100	0	0	1	100	2	100	1	100
*S. pasteuri*	0	0	1	100	0	0	0	0	0	0	0	0

## Data Availability

Not applicable.

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
