# Peer review of "Staphylococcus spp. Causatives of Infections and Carrier of blaZ, femA, and mecA Genes Associated with Resistance"

_antibiotics, 2023, doi:10.3390/antibiotics12040671_

Round 1

Reviewer 1 Report

The manuscript describes a study of 48 isolates of Staphylococcus spp. obtained in the hospital environment and evaluation of their resistance to a range of antibiotics and analysis of the genetic elements conferring these resistant phenotypes.

While the manuscript is generally well written it contains several sections where the meaning is unclear and/or include seemingly contradictory statements. Extensive revisions should be undertaken to improve the clarity of the manuscript. The Discussion section could be more focused and concentrate on the novel aspects revealed by this study.

Specific comments

Abstract lines 17-18

 This study evaluated the presence of blaZ, femA and mecA Chromosomal and plamidial genes DNA 

This study evaluated the presence of blaZ, femA and mecA chromosomal and plasmid genes

Line 21 and throughout

plasmidial  plasmid borne

Line 31

Intensive Centers unit  Intensive Care Units???

Line 34
researcher
 researchers

Line 82
located on the plasmid
 located on plasmids

Line 85-87

beta-lactamases, an extracellular enzyme, occurs through the expression of the blaZ gene, usually located in the genomic material, however it can also be present in the plasmid. 

beta-lactamases, extracellular enzymes, occurs through the expression of the blaZ gene, usually located in the genomic material, however it can also be present on plasmids

Line 102
Staphylococcus aureus S. aureus

Line 116
negative coagulase  coagulase negative

Line 118-119
the less common species was S. epidermidis (1 isolated) (Figure 1).
 the least common species were S. pasteuri aand S. epidermidis (1 each isolated) (Figure 1).

Italicise blaZ throughout

Line 124-127

100% of Staphylococcus spp. tested; and positive for 50% in S. carnosus subsp. carnosus and S. auricularis and 30% in S. haemolyticus samples.

However, none of the S. agnetis and S. aureus subsp. aureus were positive for 126 chromosomal femA.

These statements above seem contradictory, please clarify.

Table 1 – the differentiation of chromosomal and plasmid sections of this table are not clearly designated – please reformat

Line 171-172
Penicillin-resistant isolates that were also found of one of the resistance genes were: 171 S. intermedius (3) and S. pasteuri (1) chromosomal femA (Table 3).
Meaning unclear please reword

Line 176-177
PCR of DNA from non-viable cultures is possible

Line 233

specie  species

Author Response

Dear Review,

The author's attached the coverletter.

Best Regards

Lilian Carla Carneiro

Reviewer 2 Report

The comments to the authors are attached.

Author Response

Dear reviewer

I attached the coverletter.

Best Regards

Lilian Carla Carneiro

Reviewer 3 Report

I find the analysis of resistant genes in the pathogens carried in the ICU health professionals' hands interesting. The study is well presented with an extended review of the existing knowledge. 

The sentence from line 43 to line 46 is too big and difficult to understand.

I would like to ask you if there were any differences between sensitive and resistant bacteria to the studied antibiotics regarding the three examined genes.

In Tables 2 and 3 which is the percentage of resistance in relation to the total number of pathogens.

Is there a clinical conclusion regarding the results of the study?

Author Response

Dear Reviewer

I attached the coverletter

Best Regards

Lilian Carla Carneiro

Round 2

Reviewer 1 Report

The authors have addressed my previous comments.